# Identification of A Novel Class of Benzofuran Oxoacetic Acid-Derived Ligands that Selectively Activate Cellular EPAC1

**DOI:** 10.3390/cells8111425

**Published:** 2019-11-12

**Authors:** Elizabeth M. Beck, Euan Parnell, Angela Cowley, Alison Porter, Jonathan Gillespie, John Robinson, Lindsay Robinson, Andrew D. Pannifer, Veronique Hamon, Philip Jones, Angus Morrison, Stuart McElroy, Martin Timmerman, Helma Rutjes, Pravin Mahajan, Jolanta Wiejak, Urszula Luchowska-Stańska, David Morgan, Graeme Barker, Holger Rehmann, Stephen J. Yarwood

**Affiliations:** 1European Screening Centre Newhouse, University of Dundee, Biocity Scotland, Bo’Ness Road, Newhouse, Lanarkshire ML1 5UH, UK; e.beck@dundee.ac.uk (E.M.B.); a.cowley@dundee.ac.uk (A.C.); a.j.porter@dundee.ac.uk (A.P.); j.gillespie@dundee.ac.uk (J.G.); J.Robinson@dundee.ac.uk (J.R.); L.Robinson@dundee.ac.uk (L.R.); a.pannifer@dundee.ac.uk (A.D.P.); v.hamon@dundee.ac.uk (V.H.); pjones@bioascent.com (P.J.); amorrison@bioascent.com (A.M.); smcelroy@bioascent.com (S.M.); 2Department of Physiology, Feinberg School of Medicine, Northwestern University, Chicago, IL 60611, USA; euan.parnell@northwestern.edu; 3Pivot Park Screening Centre, Kloosterstraat 9, 5349 AB Oss, The Netherlands; martijn.timmerman@ppscreeningcentre.com (M.T.); helma.rutjes@ppscreeningcentre.com (H.R.); 4Structural Genomics Consortium, University of Oxford, Oxford OX3 7DQ, UK; contact@sgc.ox.ac.uk; 5Institute of Biological Chemistry, Biophysics and Bioengineering, School of Engineering and Physical Sciences, Heriot-Watt University, Edinburgh Campus, Edinburgh EH14 4AS, UK; jol03@onet.eu (J.W.); ul2@hw.ac.uk (U.L.-S.); 6Institute of Chemical Sciences, Heriot-Watt University, Edinburgh EH14 4AS, UK; dm139@hw.ac.uk (D.M.); graeme.barker@hw.ac.uk (G.B.); 7Department of Molecular Cancer Research, Centre of Biomedical Genetics and Cancer Genomics Centre, University Medical Centre Utrecht, 3508 TC Utrecht, The Netherlands; hrehman2@umcutrecht.nl

**Keywords:** cyclic AMP, EPAC, uHTS, Rap1, guanine nucleotide exchange factor (GEF) activity, agonists

## Abstract

Cyclic AMP promotes EPAC1 and EPAC2 activation through direct binding to a specific cyclic nucleotide-binding domain (CNBD) within each protein, leading to activation of Rap GTPases, which control multiple cell responses, including cell proliferation, adhesion, morphology, exocytosis, and gene expression. As a result, it has become apparent that directed activation of EPAC1 and EPAC2 with synthetic agonists may also be useful for the future treatment of diabetes and cardiovascular diseases. To identify new EPAC agonists we have developed a fluorescent-based, ultra-high-throughput screening (uHTS) assay that measures the displacement of binding of the fluorescent cAMP analogue, 8-NBD-cAMP to the EPAC1 CNBD. Triage of the output of an approximately 350,000 compound screens using this assay identified a benzofuran oxaloacetic acid EPAC1 binder (SY000) that displayed moderate potency using orthogonal assays (competition binding and microscale thermophoresis). We next generated a limited library of 91 analogues of SY000 and identified SY009, with modifications to the benzofuran ring associated with a 10-fold increase in potency towards EPAC1 over SY000 in binding assays. *In vitro* EPAC1 activity assays confirmed the agonist potential of these molecules in comparison with the known EPAC1 non-cyclic nucleotide (NCN) partial agonist, I942. Rap1 GTPase activation assays further demonstrated that SY009 selectively activates EPAC1 over EPAC2 in cells. SY009 therefore represents a novel class of NCN EPAC1 activators that selectively activate EPAC1 in cellulae.

## 1. Introduction

Exchange protein activated by cyclic AMP (EPAC) proteins are now emerging as drug targets with the potential to treat multiple disorders, including type 2 diabetes and cardiovascular diseases [1]. EPACs are multi-domain proteins that act as a guanine nucleotide exchange factor (GEF) for the Ras-like small GTPases, Rap1 and Rap2 [2,3]. The EPAC protein family comprises two members, EPAC1 and EPAC2, which are encoded by two independent genes that display distinct tissue expression; with EPAC1 ubiquitously expressed in most tissues, whereas EPAC2 expression is more restricted to the liver, brain, heart, and secretory organs, including the pancreas [4,5]. They share the same functional domain organization and mechanisms of activation, consisting of a regulatory N-terminal region, which includes a disheveled-EGL-pleckstrin homology domain (DEP) and a cyclic nucleotide binding domain (CNBD), and a catalytic C-terminal region that includes a Ras exchange motif (REM), a Ras association domain (RA), and a CDC25 homology domain (CDC25-HD) [4,5]. EPAC2 includes an additional CNBD within its N-terminus (CNBD-A) that has a low affinity for cAMP and is not required for protein activation; rather it is suggested to be involved in the subcellular localization of EPAC2 [6]. 

X-ray crystallography has provided conformations of the open [7] and closed [8] states of EPAC2 while NMR spectroscopy has been used to explore how ligand binding affects the dynamics of the conformational switch in EPAC1 [9,10,11]. In addition, we have developed an in silico docking model for interactions of the known EPAC ligand, I942 (Figure 1; [12]), with EPAC1 [13]. From these studies, it is clear that EPAC2 exists as multiple conformations in dynamic equilibrium. These range from a closed, auto-inhibited conformation in the absence of cyclic AMP, where the regulatory region sterically blocks access to the catalytic site, to a cyclic AMP bound form in an open, catalytically active state, where the regulatory region binds to a different face of the catalytic domain. Note that an EPAC1 structure has yet to be reported and in silico models have been constructed using an EPAC2 lysine 405 to glutamine point mutation structure, which is anticipated to be a roughly analogous docking site to that of EPAC1 [13].

A number of laboratories, including ours, have used high-throughput screening (HTS) of compound libraries to search for novel EPAC ligands that can discriminate between the active (agonists) and inactive (antagonists) conformations of EPAC. These assays principally use competition for binding of the fluorescent cyclic AMP analogue, 8-NBD-cAMP, to EPAC isoforms. Initially, 8-NBD-cAMP competition assays were applied to screens involving EPAC2 and led to the identification of the EPAC2-selective inhibitor, ESI-05 (4-methylphenyl-2,4,6-trimethylphenylsulfone) [14]. Furthermore ESI-09 (2-(5-(*tert*-butyl)isoxazol-3-yl)-*N*-(3-chlorophenyl)-2-oxoacetohydrazonoyl cyanide) was proposed as a non-selective EPAC1 and EPAC2 inhibitor [15,16,17], though one study showed general, unspecific protein denaturing properties of this compound at high concentrations [18]. Both uncompetitive (CE3F4) and non-competitive EPAC1 inhibitors (5225554 and 5376753) have also been identified by HTS using an *in vitro* EPAC1 guanine nucleotide exchange factor (GEF) activity assay [19] and an EPAC-based bioluminescence resonance energy transfer-based assay [20], respectively. Notably, none of these HTS approaches led to the identification of small molecule agonists of EPAC1 activity, and, to date, only cyclic AMP derived EPAC activators have been developed, independently of HTS, namely D-007 for EPAC1 and S-220 for EPAC2 (Figure 1; [21]). Recently we screened a 5195 small molecule library using HTS with competition binding of 8-NBD-cAMP to the recombinant CNBD of EPAC1 (amino acids 169-318) and identified the novel ligand, I942 [12]. Subsequent, *in vitro* EPAC1 GEF assays revealed that I942 displayed partial agonist properties towards EPAC1, leading to activation of EPAC1, in the absence of cyclic AMP, and inhibition of GEF activity in the presence of cyclic AMP, with little agonist action towards EPAC2 or protein kinase A (PKA) [12]. This was the first observation of non-cyclic-nucleotide (NCN) small molecules with agonist properties towards EPAC1. Subsequent studies with I942 revealed that it activates EPAC1 and Rap1 GTPase in cells and exerts anti-inflammatory actions in human umbilical vascular endothelial cells (HUVECs) through the inhibition of interleukin 6 (IL-6)-promoted gene expression [22].

Here, for the first time, we have adapted the 8-NBD-cAMP/EPAC1 CNBD competition assay for ultra HTS (uHTS) to explore further the chemical diversity of novel NCN EPAC1 agonists. By screening a chemically diverse library of approximately 350,000 compounds, we identified further NCN EPAC1 agonists that are chemically distinct from I942. We next synthesized an expanded analogue library from isolated hits and, with subsequent triage using binding assays and microscale thermophoresis (MST), we determined potency and selectivity of these analogues towards the CNBDs of EPAC1 and EPAC2. Subsequent *in vitro* and *in cellulae* EPAC1 activation assays led us to identify SY009 as an activator of EPAC1 that is chemically distinct from I942. The identification of further NCN agonists with the potential to activate EPAC1, independently of EPAC2, presents powerful experimental tools to investigate the role of EPAC1 in health and disease and, therefore, the development of future therapeutic strategies to combat diseases associated with EPAC activation.

## 2. Materials and Methods

### 2.1. Materials

Forskolin, rolipram, and cyclic AMP were purchased from Merck-Millipore (Burlington, MA, USA). Analogues of cyclic AMP, 8-NBD-cAMP, Sp-8-BnT-cAMPS (S-220), and 8-pCPT-2′-O-Me-cAMP (D-007) were purchased from Biolog Life Science Institute GmbH & Co. KG (Bremen, Germany). BL-21 cells were purchased from New England Biolabs. The test compound I942 (N-(2,4-dimethylbenzenesulfonyl)-2-(naphthalen-2-yloxy)acetamide) was sourced from MolPort (Riga, Latvia). Dulbecco’s modified Eagle’s medium (DMEM), fetal bovine serum (FBS), GlutaMAX, and penicillin/streptomycin (5000 U/mL) were purchased from Thermo Fisher Scientific (Waltham, MA, USA) and the selective antibiotic, puromycin, and complete, EDTA-free protease inhibitor cocktail were from Sigma-Aldrich (St. Louis, Mo, USA). Rap1A/Rap1B (26B4) and vasodilator-stimulated phosphoprotein (VASP; 9A2) antibodies were obtained from Cell Signaling Technology (Danvers, MA, USA) and HRP-conjugated anti-mouse and anti-rabbit IgG secondary antibodies from Sigma-Aldrich.

### 2.2. Recombinant Protein Production

EPAC1-CNBD (amino acids 169–318 of EPAC1) and EPAC2-CNBD (amino acids 304-453) cDNAs were previously sub-cloned into the multi-cloning site of the pGEX-6P-1 expression vector (GE Healthcare) by BC Bioscienes (Dundee, Scotland). Ral-GDS-RBD (amino acids 788-884) was cloned into pGEX-5X-1 as described [23]. Recombinant EPAC1-CNBD, EPAC2-CNBD, EPAC1 (amino acids 149–881), Ral-GDS-RBD (amino acids 788-884), and Rap1B (amino acids 1–167) glutathione S-transferase (GST)-fusion proteins were then expressed and purified from BL-21 *Escherichia coli*, as previously described [12,23]. Protein concentrations were determined by the bicinchoninic acid (BCA) assay or by Nanodrop 2000/2000c (Thermo Scientific). All recombinant proteins were determined to be pure by SDS-PAGE analysis. Proteins were stored in aliquots at −80 °C until required.

For large-scale preparations of EPAC1-CNBD for uHTS, pGEX-6P-1.EPAC1-CNBD was transformed into BL21 (DE3)-R3-pRARE2 expression strain (at Structural Genomics Consortium, University of Oxford). Cells were grown at 37 °C until the O.D_6oo_ reached between 1–1.5 and then induced with 0.1 mM isopropyl β-d-1-thiogalactopyranoside (IPTG) at 18 °C overnight. Cells were harvested next day and re-suspended in 50 mM Tris-HCl, 300 mM NaCl, 1 mM EDTA, 1 mM TCEP pH 7.8 + 0.5 mg/mL lysozyme, 1:1000 dilution of protease inhibitor cocktail, SET III and 0.1% (*v*/*v*) Triton X-100 and benzonase. The total volume after cell resuspension was 400 mL, which was split into two fractions of 200 mL each and lysed by sonication, 35% amplitude on 750 W sonicator, 5 s ON and 5 s OFF, for 10 min (total time of 20 min). The lysate was then centrifuged at 17,500× gmax on a JA18 rotor for 1 h then the supernatant was filtered through 0.45 μm filters. Equilibrated 50% (*v*/*v*) glutathione Sepharose slurry (12 mL to 300 mL filtrate) was then added to the filtrate and then rotated at 3 rpm overnight. Next day the Sepharose was washed with 60 column volumes of the wash buffer (50 mM Tris-HCl, 300 mM NaCl, 1 mM EDTA, 1 mM TCEP pH 7.8), and then eluted with 8 × 10 mL fractions of 15 mM reduced glutathione in the elution buffer (50 mM Tris-HCl, 300 mM NaCl, 1 mM EDTA, 1 mM TCEP pH 7.8). The protein fractions were run on SDS-PAGE and the eluates were buffer-exchanged overnight with 50 mM Tris-HCl, 300 mM NaCl, 1 mM EDTA, 1 mM TCEP pH 7.8, using 3.5 kD cut off snakeskin. Protein was then aliquoted into 500 μL aliquots, at a concentration of 1.4 mg/mL, and then snap frozen.

### 2.3. 8-NBD-cAMP Competition Assay and Ultra-High Throughput Screening (uHTS)

8-(2-[7-nitro-4-benzofurazanyl] aminoethylthio) adenosine-3′,5′-cyclic monophosphate (8-NBD-cAMP) competition assays were performed with purified EPAC1-CNBD and EPAC2-CNBD GST-fusion proteins as previously described [12]. Comparison of reference molecules, dose response curves (DRC) and pilot screens were done in black, low volume 384 assay plates (Greiner). Plates were incubated for 4 h before 8-NBD-cAMP fluorescence intensity at 480/535 nm (ex/em) was measured using the Envision multi-label plate reader (Perkin-Elmer; Waltham, MA, USA).

The high-throughput screen with the European Lead Factory (ELF) library (~350,000 compounds) [24,25] was carried out using the 8-(2-[7-nitro-4-benzofurazanyl] aminoethylthio) adenosine-3′, 5′-cyclic monophosphate (8-NBD-cAMP) competition assay to measure binding to the CNBD of EPAC1 as described [12]. Firstly, the competition assay was optimized to run on an operational ultra-high throughput screening (uHTS) system, comprising an Echo^®^ 555 acoustic liquid dispenser and three EnVision multi-label plate readers (Perkin-Elmer). This involved defining dispensing volumes, plate types (384 or 1536 well formats), reagent stability and incubation times. As previously reported [12], the assay was measured after 4 h, however, the feasibility assays showed a better signal to background ratio (S/B) for 400 nM probe (8-NBD-cAMP) with 400 nM protein (EPAC1-CNBD) and that this combination of reagents was stable for up to 20 h at 4 °C. The signal to background ratio (S/B) and Z’ values for the assays were comparable in both 384 and 1536 well formats. For uHTS, dose response curve (DRC) plates were prepared in 1536-well format at the European Screening Centre (Newhouse, Scotland) and shipped to the Pivot Park Screening Centre (Oss, Netherlands). The 1536-well source plates contained 100% DMSO in the first 6 columns followed by 6 compound DRCs consisting of 7 points with a final range of 2.00 × 10^−5^ M–2.00 × 10^−8^ M, resulting in a maximum of 192 compound DRCs per plate. ActivityBase High Content and Throughput Screening Software (IDBS) was used to analyze the data and to calculate quality parameters including signal to background ratio (S/B) and Z’ values. The signal to background ratio (S/B) was determined by dividing the mean maximum fluorescent signal (DMSO diluent control) by the mean minimum signal (saturating cAMP concentrations) to give the magnitude of the fluorescence change in the assay. The Z’ factor was also calculated from the minimum (*n*, cAMP) and maximum (*p*, DMSO) control values from the assay, together with the mean fluorescent values (*μ*) and their standard deviation (***σ***), using the following equation:(1)$$1-\frac{3\left(\sigma_{p}+\sigma_{n}\right)}{\left|\mu_{p}-\mu_{n}\right|}$$

For all compounds, the % effect was calculated for all seven concentrations of the DRC. ActivityBase was also used to calculate the pIC_50_, hill slope, and Emax (maximal inhibition of 8-NBD-cAMP binding, reported in Appendix A). Using optimized assay conditions, the screening of all plates demonstrated an S/B of 2.8 or higher and the Z’ value above 0.6, therefore all plates were validated, resulting in the identification of 68 actives. Of these, 23 were subsequently confirmed on a re-test of the primary assay (S/B of ±4 and a Z’ value of 0.56). Assay interference compounds were removed with a deselection assay involving interference of green fluorescent protein (60 nM GFP; excitation 480; emission 535) under regular assay conditions.

### 2.4. Bayesian Modeling

Confirmed actives from the uHTS screen were supplemented with compounds identified from an in silico Bayesian model constructed from the original screening data to identify potential false negatives. Briefly, a threshold for activity was defined based on the Z-score and % inhibition and each molecule was placed in either an active or inactive bin. All the molecules were then broken down in silico into sub-structural features. The frequency of each sub-structural feature in the active and inactive bins was then generated. A feature enriched in the active molecules was given a positive score, with a magnitude depending on the level of enrichment. Conversely, sub-structures that occurred more frequently than chance in the inactive compounds were given a negative score. The score was further weighted by how well that feature was represented in the screening data set. A table of all the substructures represented in the screening library and their associated scores was then generated. The Bayesian score does not describe how active a molecule is predicted to be. The higher score indicates a higher confidence that the molecule will be active at the threshold used to build the model but makes no prediction about it being more or less potent.

### 2.5. Generation of in Silico Docking Models of EPAC1 CNBD

While there are no published EPAC1 structures in the literature, several EPAC2 structures are available in their active form in the Protein Data Bank (PDB), co-crystallized with cyclic AMP and three different analogues, including D-007. Two of the published structures contain a mutation on position 405 (EPAC2 numbering), a lysine in EPAC2 is replaced by a glutamine (as found in the analogous position of EPAC1). The difference between the two residues K ˃ Q results mainly in a change in the local structure of the hinge region. Moreover, the difference in the cAMP analogue used in 4MGY and 4MGK (both mutated K405Q) induces a change in this region as well. Crystal structures of cAMP and these analogues bound to EPAC2 and EPAC2 mutants (K405Q) were used to guide development of the EPAC1 homology model using the Schrodinger Suite molecular modeling software platform.

### 2.6. Chemical Synthesis—General Methods

All non-aqueous reactions were carried out under oxygen free N_2_ using flame-dried glassware. Tetrahydrofuran (THF) and dimethylformamide (DMF were purified by MBRAUN SPS-800 solvent purification system. Reactions using Zn(CN)_2_ were performed under N_2_ on a Schlenk line equipped with a CuSO_4_ bubbler to trap any HCN released. Petroleum ether refers to the fraction of petroleum ether boiling in the range of 40–60 °C and was purchased in Winchester quantities. Brine refers to a saturated solution of NaOH in water. Water was distilled water. Flash column chromatography was carried out using Matrix silica gel 60 from Fisher Chemicals. Thin layer chromatography was carried out using commercially available Merck F254 aluminum-backed silica plates visualized by UV (254 nm) or stained using either a solution of acidified ninhydrin in ethanol or aqueous acidic KMnO_4_. Proton (300 or 400 MHz) and carbon (75.5 or 101 MHz) NMR spectra were recorded on Bruker AV 300 or AV 400, respectively. For samples recorded in CDCl_3_ and (CD_3_)_2_CO, chemical shifts are quoted on parts per million, relative to CHCl_3_ (δH 7.26) or (CD_3_)_2_CO (δH 2.05, central line of quintet) and CDCl_3_ (δC 77.16, central line of triplet) or (CD_3_)_2_CO (δC 29.84 using Distortionless Enhancement by Polarization Transfer (DEPT) experiments). Coupling constants (J) are quoted in Hertz. Melting points were obtained (central line of septet). Carbon NMR spectra were recorded with broad band decoupling and assigned on a Stuart Scientific SMP 10 at ambient pressure. Infrared spectra were recorded on a Perkin-Elmer Spectrum 100 FT-IR Universal ATR Sampling Accessory deposited neat to a diamond/ZnSe plate. Mass spectra were obtained at the EPSRC UK National Mass Spectrometry Facility at Swansea University.

### 2.7. Chemical Synthesis—Representative Synthesis of SY009

A chemical library of 69 analogues of the top hit, SY000, was generated (see Appendix A) following in silico modeling of the EPAC1 CNBD (described in Section 2.5). This was supplemented with 22 additional compounds obtained from the initial screening library (shaded red in the Appendix A). Synthesis of the improved hit, SY009 (illustrated in Figure 2) was done as follows. Nitrile 1 was treated with an excess of *iso*-propylmagnesium bromide, followed by treatment with aqueous acid to affect imine hydrolysis, giving phenol ketone 2 in 91% yield. Next, the benzofuran core was assembled: phenol 2 was treated with α-bromo ketone 3 in DMF in the presence of K_2_CO_3_ for 2 h at 60 °C, before a solvent exchange and refluxing in toluene for 16 h under a Dean–Stark trap in the presence of a catalytic amount of para-TsOH gave bromobenzofuran 4 in 96%. Cyanobenzofuran 5 was then obtained in 89% yield after a palladium-catalyzed cyanation using 7 mol% Pd_2_(dba)_3_ and 14 mol% dppf with 1.2 eq. Zn(CN)_2_. The reaction was carried out over 4 h in DMF at 120 °C and run under a constant stream on N_2_ on a Schlenk line equipped with a copper sulfate bubbler to capture any evolved HCN. Global reduction using LiAlH_4_ gave amino alcohol 6 in 49% yield, and re-oxidation to the aminoketone 7 using MnO_2_ was achieved in 21% yield, monitoring the reaction carefully using thin layer chromatography (TLC) to avoid oxidation of the amine to the corresponding aldimine. Repeated attempts to improve this yield using different conditions resulted in either inefficient oxidation of the alcohol, or complete aldimine formation. Next, amide 8 was formed in 89% yield after treatment of 7 with ethyl chlorooxoacetate, and subsequent ester hydrolysis gave SY009 9 in 85% yield. Routes analogous to the representative synthesis of SY009 described here were used to synthesize the other 69 compounds in the library. In each case, the appropriate phenol was selected to install the benzofuran core together with the relevant bromoketone or bromoamide to install the ketone or amide motif in the benzofuran 2-position. Well-established functional group transformations were then used to diversify groups at the benzofuran 5 position and at the carbonyl position.

### 2.8. Microscale Thermophoresis (MST)

EPAC1-CNBD and EPAC2-CNBD were labeled with the red fluorescent dye NT-647-NHS, which non-selectively couples to free amine groups in the protein, predominantly represented by exposed lysines. Briefly, 100 μL of 20 μM CNBD in PBS was incubated with 100 μL of 3-fold excess NT-647 NHS fluorescent dye for 30 min in the dark at room temperature. To remove unreacted “free” dye, the reaction was placed into a pre-equilibrated gravity flow column in MST buffer (50 mM Tris-HCl, pH 7.4, 150 mM NaCl, 10 mM MgCl_2_). The labeled protein was eluted at 3 μM in MST buffer. Red fluorescence (excitation 650 nm/emission 670–690 nm) was measured for a series of labeled CNBD concentrations to confirm that labeling was successful. A concentration-dependent fluorescent signal indicated that coupling was successful. For MST a signal > 250 fluorescent units is typically considered sufficient for screening using the Monolith NT automated instrument from NanoTemper Technologies. CNBDs were therefore used at 120 nM. 

### 2.9. Guanine Nucleotide Exchange Factor (GEF) Activity Assay

The EPAC1 *in vitro* GEF activities, using recombinant EPAC1 (149–881) and Rap1B (1–167), were calculated as previously described [12]. Briefly, recombinant EPAC1 (149–881) was incubated at 100  nM with recombinant Rap1B (200 nM) preloaded with the fluorescent guanosine diphosphate (GDP) analogue 2′/3′-O-(N-Methylanthraniloyl)guanosine 5-diphosphate (MANT-GDP) in the presence of 20 μM GDP and then the fluorescence intensity was measured at 360/450 nm (ex/em) over time. Multiple reactions were performed at cyclic AMP/compound concentrations varying between 0.1 and 1000 μM and curves were fit as single exponential decay to obtain k_obs_. 

### 2.10. Active Rap1 Pull-Down Assay

U2OS cells stably transfected with EPAC1 or EPAC2 (from Holger Rehmann, University of Utrecht) were cultured in 6-well plates in DMEM, high glucose, supplemented with 10% (*v*/*v*) FBS, 1% (*v*/*v*) GlutaMAX, 1% (*v*/*v*) penicillin-streptomycin, and 2 mg/L puromycin (to ensure selection of stable transfectants). Then, 80% confluent cells were starved in culture medium with reduced FBS concentration (0.5% *v*/*v*) for 16 h and then stimulated for 10 min with either vehicle, 100 µM of test compounds, 10 µM forskolin, and 10 µM rolipram or 50 µM of D-007 in case of U2OS-EPAC1 or 100 µM of S-220 for U2OS-EPAC2. Cells were then rinsed with ice-cold PBS and lysed in 0.5 mL cell lysis buffer (from Cell Signaling Technologies) supplemented with 10 mM MgCl_2_ and 1 mM phenylmethylsulfonyl fluoride (PMSF), followed by clearing the lysates by centrifugation. Cell lysates were incubated for 1 h (4 °C, gentle agitation) with 40 µg GST-Ral-GDS-RBD immobilized on glutathione Sepharose 4B (GE Healthcare) to selectively capture GTP-bound Rap1. Following this, the glutathione resin was separated from supernatant by centrifugation, washed three times with cell lysis buffer, then resuspended in 2× SDS sample loading buffer and denatured for 5 min at 95 °C followed by analysis by Western blotting.

### 2.11. SDS-PAGE and Western Blotting

Samples were prepared by mixing equal volumes of cell lysate with 2× SDS sample loading buffer and then denaturing for 5 min at 95 °C, unless indicated otherwise. Protein samples were separated by SDS-PAGE on 10% (*v*/*v*) polyacrylamide gels, for VASP, or on 12.5% (*v*/*v*), for Rap1, and then transferred to nitrocellulose membranes. Membranes were then blocked for 1 h at room temperature in 5% (*w*/*v*) bovine serum albumin (BSA) in Tris-buffered saline containing 0.1% (*v*/*v*) Tween 20, followed by an overnight incubation with primary antibody diluted in blocking buffer at 4 °C. Subsequently, the membranes were incubated with appropriate horseradish peroxidase-conjugated secondary antibodies for 1 h at room temperature. For signal detection the SuperSignal West Pico PLUS Chemiluminescent Substrate (Thermo Scientific) was used. Images were acquired using the Fusion FX7 camera platform (Vilber; Collégien, France). Densitometry was performed with ImageJ.

### 2.12. Statistical Analyses

Statistical significance was determined using one-way analysis of variance (ANOVA) with Tukey’s post-tests. 

## 3. Results

### 3.1. Ultra-High-Throughput (uHTS) Screening to Identify Interacting Ligands for the EPAC1 CNBD

The high-throughput screen with the European Lead Factory (ELF) library (~350,000 compounds) was carried out using our previously reported 8-NBD-cAMP competition assay to measure binding to the CNBD of EPAC1 [12] under optimized conditions, as described in Section 2. During screening all plates showed an S/B of 2.8 or higher and a Z’ value above 0.6, therefore all plates were validated. This resulted in the identification of 68 actives, 23 of which were subsequently confirmed on a re-test of the primary assay (S/B of ±4 and a Z’ value of 0.56). Assay interference compounds were removed with a deselection assay (GFP interference, as described in Section 2).

Confirmed actives from the uHTS screen were supplemented with compounds identified from a Bayesian model constructed from the original screening data to identify potential false negatives. Dose response curves were then determined with the primary EPAC1-CNBD binding assay (Table 1). Only three compounds returned a pEC_50_ > 5 with an acceptable Hill slope. As such all compounds that demonstrated > 20% effect were selected for testing using microscale thermophoresis (MST; as described in Section 2) as an orthogonal biophysical assay to confirm target engagement [26,27,28]. The three compounds that returned a measurable pEC_50_ through the competition-binding assay, plus compounds which appeared to bind in the MST experiment and their structural analogues, were combined to give a preliminary hit list of 31 compounds. Compounds that failed liquid chromatography–mass spectrometry (LC-MS) analysis were removed. In addition, compounds that demonstrated “frequent hitter” characteristics were eliminated, and this resulted in a “qualified hit list” (QHL), after legal clearance of eight compounds. The compounds on the QHL were split into structural clusters. Overall, five clusters were identified of which three were singletons. Of these, four representative compounds were further verified by resynthesis, hit confirmation, and more extensive profiling (Table 1). Cluster 1 compounds (ECS1000851(SY000) and ESC1000906) confirmed activity in both competition binding and MST assays with good alignment to the data generated from the original screen (see Section 2.3) and displayed favorable in silico physical properties (Table 2). 

### 3.2. Development of An in Silico Docking Model of EPAC1

Due to the similarities in the two hit structures (Table 2), further structure activity relationship (SAR) investigation was carried out on SY000 (Figure 3A) with a focus on confirming that further analogues demonstrated responsive SAR, by probing key binding interactions and improving EPAC1 potency. Using the Schrodinger Suite to carry out in silico docking to a model of the EPAC1 CNBD (Figure 3B), revealed that the oxoacid functional group of SY000 interacts with the region of the EPAC1 CNBD that normally binds the phosphate group of cylic AMP, through an H-bond network with at least three of the potential H-bond donors in the same region (Figure 3B). The rotational degree of freedom of the oxoacid moiety, relative to the benzofuran core is quite high, allowing for an adaptability in the position during the docking. This leads to several possible options for the binding mode and a relatively high degree of uncertainty in the poses selected. Given the high degree of uncertainty in the poses selected, SAR studies were aimed at probing potential interaction points and expanding into novel trajectories to establish key binding elements of SY000.

### 3.3. SAR Analysis of Structural Analogues of SY000 with the EPAC1 CNBD

Preliminary SAR on a benzofuran oxo-acetic acid series, based on SY000, was done using 8NBD-cAMP binding to EPAC1 and EPAC2 CNBDs, as summarized in the Appendix A and Figure 4. Initial SAR indicated that the R1 acid moiety is important for binding activity to both EPAC1-CNBD and EPAC2-CNBD, with ethyl ester, amide or ketone replacement groups ablating binding activity (Appendix A). Further alternatives to the R1 group (Appendix A), including reducing the linker length by a single methylene (ESC1000915), further variations in linker length (ESCs 1001634, 1001650, 1001696, and 1001702), and replacing the oxo-acid with isosteres (ESCs 1001000, 1001622, 1001792, and 1001918) all ablated binding activity, indicating that the spatial relationship between the acidic functional group and the aromatic core is important for interaction with EPAC1 and EPAC2. In addition, *N*-methylation of the amide portion (ESC100960) also results in loss of activity. This could indicate a role for the NH as a hydrogen bond donor or that the substitution could adversely affect the conformation of the adjacent oxo-acetic acid group.

Extension from other trajectories of the aromatic core were also investigated. ESC1001955 indicates that methoxy substitution at the R5 position is well tolerated at both EPAC1 and EPAC2, although the analogous NH_2_ substitution is detrimental to activity (Appendix A). In contrast, methoxy substitution at the R4 position is not tolerated (Appendix A). It is likely that substitution at R4 adversely affects the conformation of the tethered oxo-acetic acid moiety at R1. This idea is supported by the fact that insertion of the oxo-acetic acid tail at R4, rather than R1, in ESC1001896 is tolerated at EPAC2 but detrimental to EPAC1 binding (Appendix A).

Given the importance of the relationship between the oxo-acetic acid tail and the core/aryl ketone moiety (R7), we were interested in addressing whether alternative 5–6 heterocyclic systems could also act as suitable core replacements (Appendix A). We found that changes to the core resulted in a dramatic loss of EPAC1 and EPAC2 binding activity, however it was interesting to note that the *N*-benzyl indole core (ESC1001264 and ESC1001466) maintained some EPAC2 binding activity (Appendix A). A small set of analogues was also prepared to probe the importance of the ketone as a linker (R6) to the core by inserting modified substituents at the R2 position (Appendix A). ESC1001581 indicates that replacing the ketone carbonyl with a simple methylene linker results in ~10-fold reduction in binding activity towards the EPAC1 CNBD. Interestingly, the same analogue maintained some EPAC2 binding activity (~5-fold reduction in potency towards EPAC2). Analogues ESC1001536 and ESC1001547 also indicate that replacing the carbonyl with an alcohol is more detrimental to EPAC1 binding than EPAC2.

Insertion of other substituents at the R2 position, indicates that aryl and heteroaryl ketone substituents are tolerated, with the exception of pyridyl ketones (Appendix A), suggesting that the aryl moiety may point out towards to the CNBD solvent channel, with further extension having limited benefit. The relatively flat SAR with aryl ketones at the R2 position led us to speculate that non-aromatic replacements may be feasible at this position. Indeed, alternatives such as *c*-hexyl and adamantyl were well tolerated but had a tendency to favor interactions with EPAC2 over EPAC1 (Appendix A). Small substituents, such as *i*-propyl, resulted in a modest decrease in potency but ligand efficiency was maintained. Introduction of a more polar tetraydropyran also resulted in a modest decrease in binding at both EPAC1 and EPAC2. Amide substitutions at R2 were also generally unfavorable towards EPAC1 (Appendix A).

A range of three substituted benzofuran analogues (R3) was also prepared and tested against EPAC1 and EPAC2 CNBDs. Many of the R3 substituents tested were tolerated at both the EPAC1 and EPAC2 CNBDs (Appendix A). For EPAC2, there was a noticeable enhancement of potency with small (methyl and ethyl) substituents, however further extension with bulkier aliphatic or aromatic groups proved to be deleterious. For EPAC1, ethyl and *i*-propyl were optimal substituents with larger groups (*c*-hexyl and phenyl) being poorly tolerated. For both EPAC1 and EPAC2, O-alkyl substituents were not well tolerated, indicating a preference for lipophilic substituents or suggests an unfavorable alteration to the conformation of the adjacent ketone position. The analogue ESC1001866 (further referred to as SY009) was identified as a lead compound from this series as demonstrating ~10-fold binding selectivity for EPAC1 over EPAC2 (Appendix A) while maintaining similar ligand and lipophilic efficiency to SY000 (see Appendix A in accompanying EXCEL file). In addition, SY009 demonstrated enhanced (~10-fold) binding potency when compared to SY000 and the known NCN partial agonist, I942 (Figure 5). This initial SAR work demonstrates that it is possible to increase binding activity and selectivity of SY000 analogues, while maintaining physiochemical properties.

### 3.4. Binding/Activity Relationships of R3 Substituted SY000 Analogues

Given the improvement of binding potency and EPAC1 selectivity of SY009 over SY000 (Figure 5), we next investigated the binding/activity relationships of two other R3-substituted ligands, SY006 and SY007 (Figure 6), which also displayed improved (~10-fold) binding over SY000 (Figure 6). We first tested the ability of SY006, SY007, and SY009 to activate EPAC1 enzyme activity in *in vitro* GEF assays (Figure 7). The assay is based on the EPAC1-stimulated dissociation of fluorescent MANT-GDP from recombinant Rap1 in the presence of excess non-fluorescent nucleotide, as described in Section 2.

We first investigated agonistic and antagonistic properties of SY006, SY007, and SY009 (Figure 7A). Activity assays were next carried out using 50 μM SY006, SY007 or SY009 in the presence of 50 μM or 500 μM cyclic AMP (Figure 7A). Under these conditions, we found that, in all cases, ligands induced EPAC1 activity in the absence of cyclic AMP, but inhibited cyclic AMP-induced EPAC1 activity as previously reported for I942 [12]. The antagonistic properties of SY006 seemed stronger than those of SY007 and SY009. This indicates that, as previously reported for I942, SY006, SY007, and SY009 are partial agonists towards EPAC1. Next, an in-depth analysis of SY009 was performed by determining the dependency of EPAC1 activity on various concentrations of SY009 (Figure 7B) in comparison with cyclic AMP and I942. We found that SY009 activated EPAC1 with a maximal activity lower than that of cylclic AMP (Figure 7B). As problems with solubility were found with higher concentrations of SY009 (as with SY006 and SY007) saturating concentrations could not be reached, which makes the exact determination of the maximal activity difficult.

Next, we used in-cell assays and immunoblotting with anti-VASP, PKA-phosphosubstrate antibodies, in cells expressing either EPAC1 or EPAC2, to determine (1) the ability of R3-substituted analogues to promote cellular EPAC activity; (2) the selectivity of analogues for EPAC1 or EPAC2; and (3) if analogues induce cellular PKA activity (Figure 8). As a positive control, EPAC1- or EPAC2-expressing cells were stimulated with a combination of forskolin and rolipram (F/R), to elevate global cyclic AMP levels through simultaneous inhibition of cellular type 4 cyclic AMP phosphodiesterase activities and activation of adenylyl cyclase (Figure 8). In addition, as a further control, EPAC1 cells were stimulated with D-007, to selectively activate EPAC1, and EPAC2-expressing cells were stimulated with S-220, to selectively activate EPAC2 (Figure 8).

Treatment with F/R, D-007 or S-220 was found to significantly activate Rap1 activity in both cell types, whereas only F/R promoted VASP phosphorylation through PKA activation (Figure 8). Treatment of cells with the improved hits, SY006, SY007 or SY009, was found to significantly activate Rap1 GTPase in cells expressing EPAC1, but not EPAC2 (Figure 8). In addition, none of the ligands were found to promote PKA activation, as determined by phospho-VASP blotting (Figure 8). Moreover, since ligands do not activate Rap1 or PKA in EPAC2 expressing cells, this indicates that they do not promote non-specific elevations in intracellular cyclic AMP levels through off-target activation of adenylyl cyclase or inhibition of endogenous cyclic AMP phosphodiesterases (PDEs). 

## 4. Discussion

Screening against EPAC1 CNBD with the ELF library (~350,000 compounds) and subsequent triage and re-synthesis identified a single compound, SY000, which confirmed moderate binding activity at EPAC1. SY000 was prioritized for further work and initial SAR resulted in compound SY009 with improved activity (sub-μM). SY009 was found to display agonist properties towards EPAC1 *in vitro* and *in cellulae* and selectivity towards EPAC1, over EPAC2 and PKA, in cells. Further work will be required to determine binding to other cellular CNBDs, including cyclic nucleotide-gated (CNG) and hyperpolarization-activated cyclic nucleotide–gated (HCN) channels. Despite this, SY009 is an encouraging starting point for further work since it has been demonstrated here that it is possible to increase the binding activity of ligands with small structural alterations, as well as induce bias towards either EPAC1 or EPAC2 activity. 

The benzofuran core is typically not a favored motif in drug discovery due to the potential susceptibility to hepatic oxidation by CYP enzymes, giving rise to reactive epoxide intermediates. In the case of this series, the C2/C3 substituents would be expected to reduce the susceptibility to epoxidation, however we were also interested in addressing whether alternative 5–6 heterocyclic systems could act as suitable replacements. We found that replacing the core benzofuran (Appendix A) resulted in a dramatic loss in EPAC1 activity. While no promising alternative cores were discovered there is scope for more extensive investigation in this area. Poor solubility, presumably due to the flat aromatic nature of the hit compounds, was identified as an important area for improvement. To address this potential limitation, analogues with reduced aromaticity were also targeted. This resulted in several analogues (e.g., ESC1001920 and ESC1001653) with equivalent potency to the original hits and significantly higher fraction sp^3^ which should help to improve solubility issues with the series.

SY006, SY007, and SY009 proved to be effective activators of Rap1 in cells and therefore complement our original NCN EPAC1 agonist, I942, as chemically distinct tool molecules to probe the role of EPAC1 in cellular systems. The question remains as to the mechanisms by which the new SY series and the original I942 series promote EPAC1 activation. We know that cyclic AMP binds the CNBD leading to subtle conformational rearrangements within the core CNBD domain in the auto-inhibited closed state, this leads to “tumbling” of the CNBD, freeing EPAC to open towards the active state, and allowing the catalytic subunit to interact with Rap GTPases. In this conformation, cyclic AMP forms additional interactions, stabilizing the CNBD in the active state. As a result, EPAC exists in dynamic equilibrium between open and closed states and, with the active state being selected upon interaction with orthosteric agonists. Partial agonists, such as I942, and SY009 identified here, are less efficient than cyclic AMP in shifting the conformational equilibrium. Despite this, the work here provides further evidence that NCN EPAC1 agonists can be further developed as tool molecules with the potential for development as future therapeutic agents. In this regard SY009, being chemically distinct from I942, is predicted to adopt a distinct binding mode in the cyclic AMP binding pocket (Figure 4). Indeed, our current model of I942/CNBD interaction involves the acidic *N*-acylsulfonamide motif (pKa ~4) occupying a similar volume to the cyclic AMP phosphate, engaging a key charge-pairing arginine (Arg279) within the CNBD phosphate-binding cassette [13]. Similarly, based on the docking model presented here the oxoacid functional group of SY006, SY007, and SY009 is also predicted to interact with the same region of the EPAC1 CNBD, in a similar fashion. However, whereas the benzofuran oxo-acetic series are predicted to orientate in the pocket in a manner analogous to cyclic AMP (Figure 3b), I942 may form additional, hydrophobic interactions that are not accessible to cyclic AMP involving threading of the oxymethylene linker through a narrow, solvent-filled passage towards the back of the protein [13]. Therefore, further comparative binding studies are required to fully define the binding modes to determine the molecular basis of selectivity and further develop potency of the two existing classes of NCN EPAC1 agonists.

## 5. Conclusions

For the first time, we have adapted the 8-NBD-cAMP/EPAC1 CNBD competition assay for uHTS to explore further the chemical diversity of novel NCN EPAC1 agonists. By screening a chemically diverse library of around 350,000 compounds, we identified further NCN EPAC1 binders that are chemically distinct from I942. Initial SAR work demonstrates that it is possible to increase binding activity, whilst maintaining or improving the physicochemical properties, of these ligands. Additionally, some compounds were found to exhibit selectivity for EPAC1 over EPAC2 and vice versa. The initial responsive SAR observed with these analogues suggests further improvements could be made and this series has potential for further lead optimization efforts. Subsequent *in vitro* and in celluae EPAC1 activation assays led us to identify SY009 as a selective activator of EPAC1 that is chemically distinct from I942. This represents a powerful experimental tool to investigate structure/function properties of EPAC1 in health and disease and, therefore, the development of future therapeutic strategies associated with EPAC activation.

## Figures and Tables

**Figure 1 cells-08-01425-f001:**
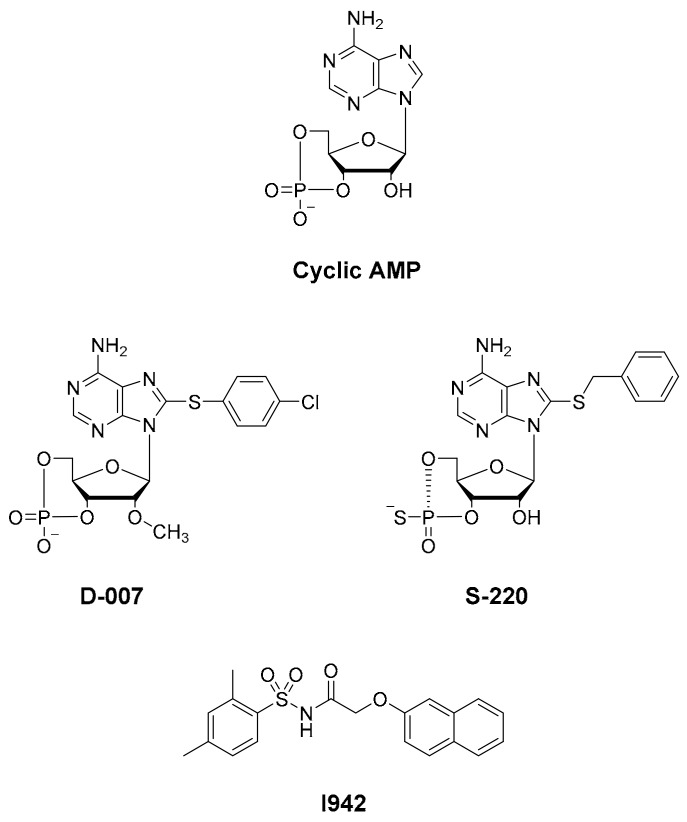
Chemical structures of existing exchange protein activated by cyclic (EPAC) activators.

**Figure 2 cells-08-01425-f002:**
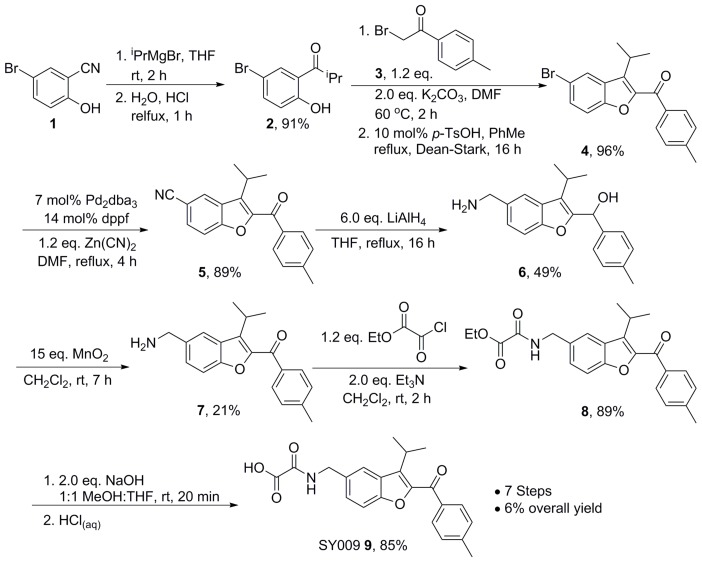
Representative synthesis of the EPAC1 activator, SY009.

**Figure 3 cells-08-01425-f003:**
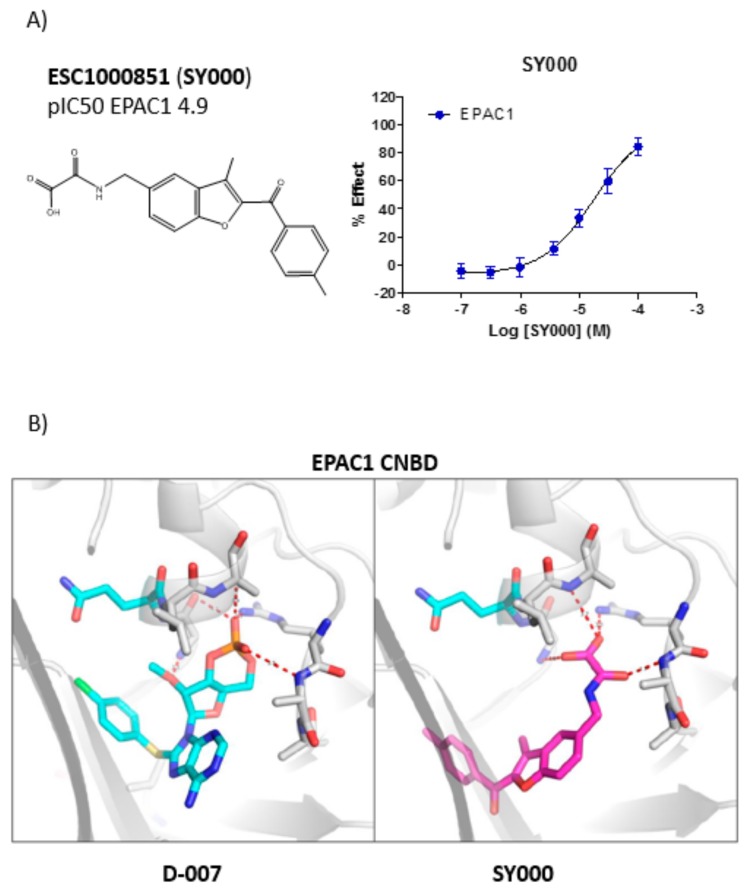
(**A**) Dose response testing of the verified hit, SY000, against the cyclic nucleotide-binding domain (CNBD) of EPAC1. The hit compound, SY000, isolated from the European Lead Factory (ELF) library was tested in a seven-point dose response format in the EPAC1-CNBD, 8-NBD-cAMP binding assays (**right**). In the graph, “% effect” refers to % inhibition of 8-NBD-cAMP binding to the EPAC1-CNBD with increasing doses of SY000. The structure of SY000 and pIC_50_ values are shown on the left. (**B**) Example pose of cyclic AMP analogue, D-007 (**left**), and SY000 (**right**) bound to an EPAC1 homology model (based on PDB 4HMO).

**Figure 4 cells-08-01425-f004:**
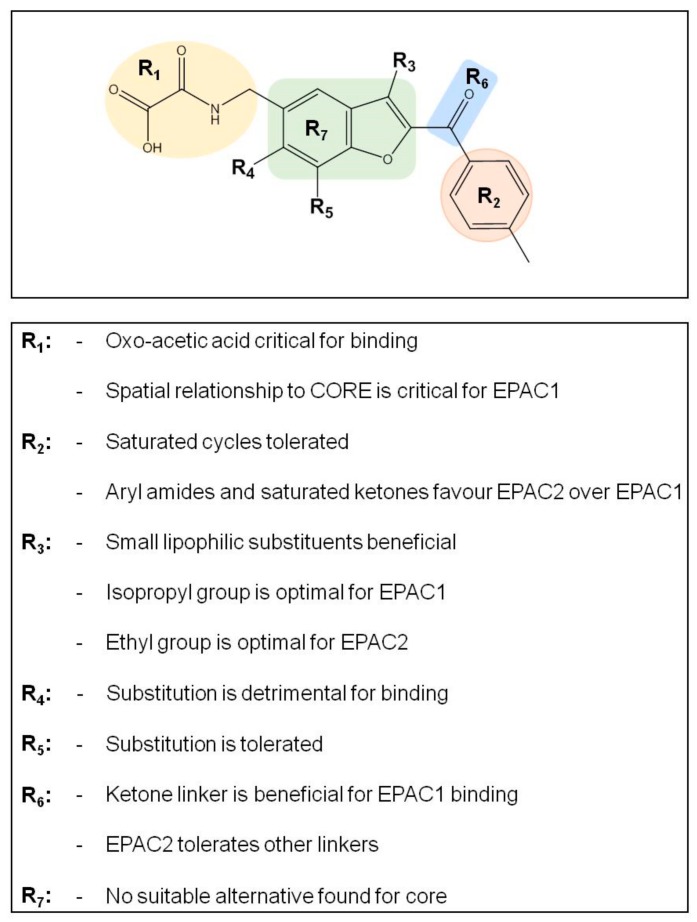
Summary of initial structure activity relationship (SAR) results for the oxo-acetic acid series. EPAC1-CNBD binding activity was confirmed through orthogonal assays (competition binding and microscale thermophoresis (MST); see Supplementary Material).

**Figure 5 cells-08-01425-f005:**
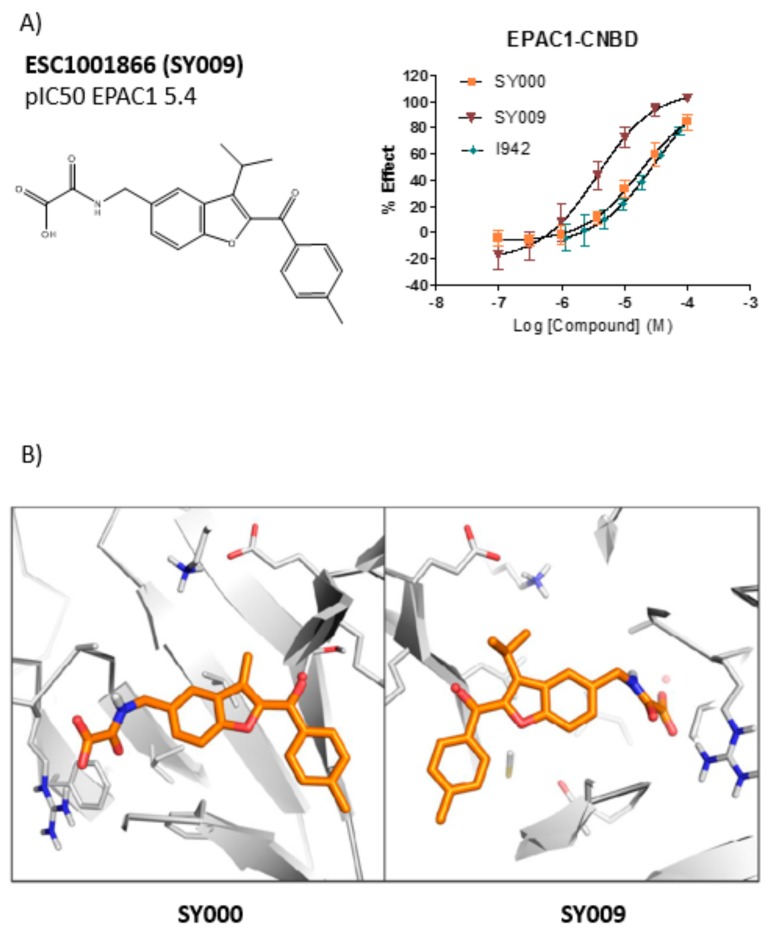
(**A**) Inhibition of 8-NBD-cAMP fluorescence, resulting from interaction with EPAC1 CNBD as described in Section 2, is plotted in the presence of varying concentrations of SY000, SY009, and the previously identified EPAC1 partial agonist, I942 [12] (*n* = 3). In the graph, “% effect” refers to % inhibition of 8-NBD-cAMP binding to the EPAC1-CNBD with increasing doses of test ligands. (**B**) Example pose of SY000 and SY009 bound to an EPAC1 homology model.

**Figure 6 cells-08-01425-f006:**
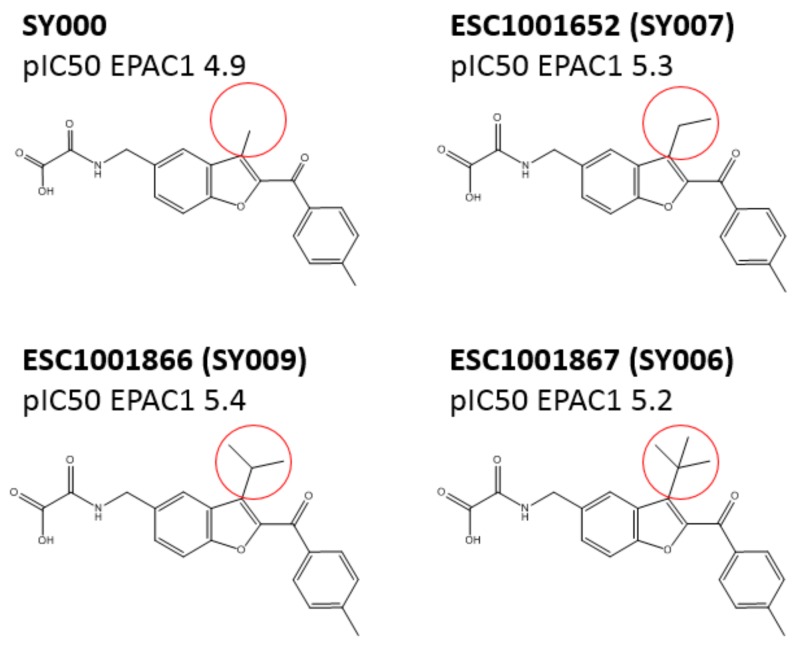
The figure indicates a range of R3 substituents (indicated by red circles) tolerated at EPAC1 CNBD. Ethyl and *i*-propyl were found to be optimal substituents. *O*-alkyl substituents were not well-tolerated.

**Figure 7 cells-08-01425-f007:**
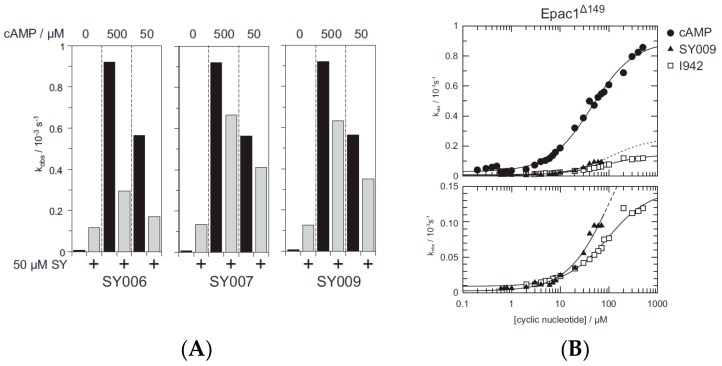
*In vitro* guanine nucleotide exchange factor (GEF) activity of EPAC1 in the presence of optimized EPAC1 ligands. (**A**) Demonstrates the rate constants from exchange reactions as bar graphs with either 500 μM or 50 μM cyclic AMP, in the presence or absence of 50 μM SY006, SY007 or SY009, respectively. (**B**) Demonstrates exchange activity induced by cyclic AMP, I942 or SY009. A plot is shown that demonstrates the dose dependency of k_obs_ on the concentration of cAMP (open circles), SY009 (closed triangles) or I942 (open squares).

**Figure 8 cells-08-01425-f008:**
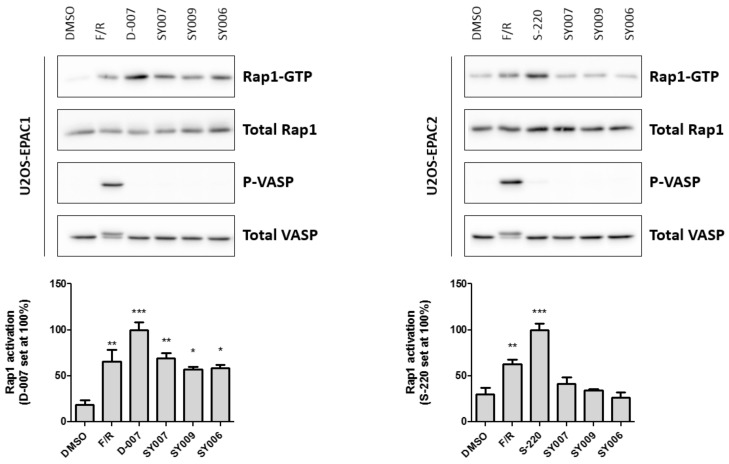
Comparison of Rap1 and protein kinase A (PKA) activation in U2OS cell lines stably expressing either EPAC1 (**left**) or EPAC2 (**right**). Cells were stimulated with either vehicle (DMSO) or a combination of 10 μM forskolin plus 10 μM rolipram (F/R), to elevate intracellular cyclic AMP levels, EPAC-selective cyclic AMP analogues D-007 (for EPAC1 cells) or S-220 (for EPAC2 cells) or the improved hits, SY006, SY007 or SY009, as indicated. The activation of PKA was monitored by a phosphorylation-induced band shift of VASP and with an anti-phospho-specific anti-VASP antibody. Rap1 GTP was precipitated from cell lysates and compared to the total Rap1 levels as described in Section 2. Quantification of Western blots is indicated as bar charts below. Results are presented as mean ± SEM with significant increases in Rap1 GTP, relative to vehicle-stimulated cells as indicated; * *p* < 0.05, ** *p* < 0.01 and *** *p* < 0.001, respectively (*n* = 3 for EPAC1; *n* = 4 for EPAC2).

**Table 1 cells-08-01425-t001:** The table below details the original potency obtained from ultra-high-throughput (uHTS) screening versus the potency with freshly synthesized solid compounds. Only the cluster 1 compounds confirmed EPAC1 binding activity in the 8-nucleotide-binding domain (NBD)-cAMP competition assay.

Cluster	ID Codes	EPAC1 (uHTS pIC_50_)	MST (pK_d_ )	EPAC1 (Resynthesis pIC_50_)
Cluster 1	ESC1000851 (SY000)	<4.7	5.5	4.9
Cluster 1	ESC1000906	<4.7	6.5	4.5
Cluster 2	ESC1000583	5.4	<4.5	<4
Singleton 3	ESC1000505	<4.7	<4.5	<4

**Table 2 cells-08-01425-t002:** The table below shows in silico calculated physical properties (using Vortex software; Dotmatics) for the two confirmed “qualified hit list” (QHL) hits. Abbreviations: MW, molecular weight; LE, ligand efficiency; LiPE, lipophilic efficiency; clogD, calculated measurements of lipophilicity; PSA, polar surface area; Fsp^3^, fraction sp^3^.

Compound Name	Structure	EPAC1 (pIC_50_)	MW	LE	LipE	cLogD	PSA	Fsp^3^
ESC1000851(SY000)	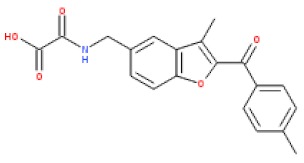	4.9	351	0.26	2.6	2.2	97	0.15
ESC1000906	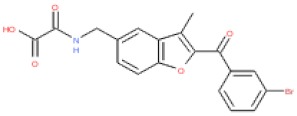	4.5	416	0.24	2.0	2.5	97	0.11

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
