# Peer review of "Identification of A Novel Class of Benzofuran Oxoacetic Acid-Derived Ligands that Selectively Activate Cellular EPAC1"

_cells, 2019, doi:10.3390/cells8111425_

Round 1

Reviewer 1 Report

The paper by Beck et al deals with the discovery of a new class of EPAC1 activators by means of HTS. The topic is interesting, it deserves publication, but the paper in the present form is very difficult to read, and for this reason I ask for a major revision.

I am not familiar with HTS, so I may not completely understand all the required steps, which are explained in the experimental part and then reported in a condensed way in the result section. I realize that the procedure may be straightforward for the expert personnel, but not for other non-expert readers. For instance, what is the meaning of Z’ value (line 168)? Is Z (line 172) the same as Z’? Sometimes just a small change can improve readability: for instance, removal of “Table 1” from lines 289-290 (or replacement with “see section 2.3) could help the reader to understand that what is reported in table 1 does not refer to the three compounds mentioned at page 7 but to the 5 clusters (even if only 3 are shown) explained at page 8.

The synthetic pathway to obtain SY009 is explained in details (Figure 1, section 2.6) but the physicochemical/spectroscopic property of the final compound and synthetic intermediates are not provided. In addition, it is not clear to me if the library of 90 compounds mentioned at line 211 is made of molecules already present in the initial set (350000 items) or compounds which have been newly synthesized. In this second instance, some property should be reported also for these substances, which couldn’t be all synthesized in the same way as SY009.

In the discussion about SAR, and also in the excel table, selectivity over EPAC2 it mentioned, however how this has been measured is not reported in the paper. EPAC2 tests are only reported in the text for 3 compounds (fig. 7), stating that the selected compounds do not activate EPAC2. However, the supplementary table shows a precise value (column D) at each row, indicating that it has been calculated for all compounds. It is mentioned in the text (lines 350 and 359) that the selectivity of SY009 is reported in fig. 4 but it is not.

SAR are discussed in a rather superficial way. First of all, since the authors differentiate SAR (section 3.2) from Binding/Activity Relationships (Section 3.3), which activity have they considered for SAR studies? Then, the authors should deepen the description of the structural modifications that they have considered to make their analysis and the outcome. For instance, only one R4 group has been inserted; the statement “ketone linker is beneficial” is not representative of the structural modification taken into account, something on the aromatic substitution on R2 could be mentioned, and so on. In my opinion, the authors should describe how they have reasoned to draw the conclusions reported in figure 3 and in the discussion.

Adenyl cyclase, PKA and PDE are not the only proteins which bind cAMP; also CNG and HCN channels have a CNBD and could be in principle modulated by EPAC activators binding to CNBD. The problem of selectivity toward these proteins should be addressed in the introduction and/or discussion.

Other remarks

Sometimes numbers, although very similar, are not the same between text, figures and supplementary file. For instance: the pEC50 for SY000 is 4.8 in table 1, 4.9 in fig. 2, 4.85 in the excel table. Please uniform.

Please insert a figure with the structure of the compounds cited in the introduction (at least the ones which are mentioned also in other part of the paper).

Legend of figure 2: most part of it should be moved to the text.

Figure 4: please insert (A) at the beginning of the legend. I suggest to show the two compounds in the same orientation, to make the comparison easier, and to highlight also the main interactions.

Lines 171-173: please check the sentence; probably the main verb is missing.

Please clarify HWU (line 190).

Line 212: the supplementary material does not describe the procedure to obtain the EPAC1 CNBD model. This should be reported somewhere.

Line 225-226: Did the authors use oxalyl chloride or ethyl chlorooxoacetate (figure 1)?

Line 314: 1B should be 2B; the same at line 317.

Line 321: “… this oxygen”: of the 5 preset in the molecule, or of the 3 in the oxalate portion, which oxygen atoms are the authors talking about?

Line 409: Fig.6 should be Fig. 7.

In the list of abbreviations, several items are not mentioned in the text (ASAP, HRMS, LRMS, AlogP, HBA, HBO).

Author Response

WE THANK THE REVIEWER FOR THEIR VERY USEFUL COMMENTS. WE HAVE INCLUDED REPLIES TO EACH SUGGESTION BELOW.

I am not familiar with HTS, so I may not completely understand all the required steps, which are explained in the experimental part and then reported in a condensed way in the result section. I realize that the procedure may be straightforward for the expert personnel, but not for other non-expert readers. For instance, what is the meaning of Z’ value (line 168)? Is Z (line 172) the same as Z’? Sometimes just a small change can improve readability: for instance, removal of “Table 1” from lines 289-290 (or replacement with “see section 2.3) could help the reader to understand that what is reported in table 1 does not refer to the three compounds mentioned at page 7 but to the 5 clusters (even if only 3 are shown) explained at page 8.

WE HAVE MODIFIED THE METHODS SECTION TO CLARIFY THE USE OF Z' AND MADE THE OTHER CHANGES THE REVIEWER SUGGESTS.

The synthetic pathway to obtain SY009 is explained in details (Figure 1, section 2.6) but the physicochemical/spectroscopic property of the final compound and synthetic intermediates are not provided. In addition, it is not clear to me if the library of 90 compounds mentioned at line 211 is made of molecules already present in the initial set (350000 items) or compounds which have been newly synthesized. In this second instance, some property should be reported also for these substances, which couldn’t be all synthesized in the same way as SY009.

WE APOLOGISE FOR THE LACK OF CLARITY. THE 90 COMPOUNDS ARE A MIX OF NEWLY SYNTHESISED ANALOGUES AND COMPOUNDS OBTAINED FROM THE ORIGINAL SCREENING LIBRARY. WE HAVE MODIFIED THE METHODS TO HIGHLIGHT THIS AND INDICATED THE CHOSEN LIBRARY COMPOUNDS IN THE SUPPLEMENTARY TABLE BY SHADING THEM RED. WE HAVE ALSO INCLUDED A SUPPLEMENTARY FILE DETAILING SPECTROSCOPIC PROPERTIES OF SY009 AND INTERMEDIATES.

In the discussion about SAR, and also in the excel table, selectivity over EPAC2 it mentioned, however how this has been measured is not reported in the paper. EPAC2 tests are only reported in the text for 3 compounds (fig. 7), stating that the selected compounds do not activate EPAC2. However, the supplementary table shows a precise value (column D) at each row, indicating that it has been calculated for all compounds. It is mentioned in the text (lines 350 and 359) that the selectivity of SY009 is reported in fig. 4 but it is not.

WE HAVE NOW INCLUDED FULL DATA FOR EPAC2 BINDING AND ACCOMPANYING DISCUSSION OF SAR.

SAR are discussed in a rather superficial way. First of all, since the authors differentiate SAR (section 3.2) from Binding/Activity Relationships (Section 3.3), which activity have they considered for SAR studies? Then, the authors should deepen the description of the structural modifications that they have considered to make their analysis and the outcome. For instance, only one R4 group has been inserted; the statement “ketone linker is beneficial” is not representative of the structural modification taken into account, something on the aromatic substitution on R2 could be mentioned, and so on. In my opinion, the authors should describe how they have reasoned to draw the conclusions reported in figure 3 and in the discussion.

WE HAVE ADDED A MUCH MORE DETAILED DESCRIPTION OF SAR AND INCLUDED THE ASSOCIATED STRUCTURAL MODIFICATIONS AS SUPPLEMENTARY FIGS.

Adenyl cyclase, PKA and PDE are not the only proteins which bind cAMP; also CNG and HCN channels have a CNBD and could be in principle modulated by EPAC activators binding to CNBD. The problem of selectivity toward these proteins should be addressed in the introduction and/or discussion.

WE HAVE NOTED THAT FURTHER WORK WILL NEED TO BE DONE TO ASSESS INTERACTIONS WITH OTHER CYCLIC AMP BINDING DOMAINS.

Other remarks

Sometimes numbers, although very similar, are not the same between text, figures and supplementary file. For instance: the pEC50 for SY000 is 4.8 in table 1, 4.9 in fig. 2, 4.85 in the excel table. Please uniform.

WE HAVE MADE THE SUGGESTED CHANGES.

Please insert a figure with the structure of the compounds cited in the introduction (at least the ones which are mentioned also in other part of the paper).

WE HAVE INCLUDED THE REQUESTED FIGURE IN THE INTRODUCTION.

Legend of figure 2: most part of it should be moved to the text.

WE HAVE MADE THE APPROPRIATE MODIFICATION.

Figure 4: please insert (A) at the beginning of the legend. I suggest to show the two compounds in the same orientation, to make the comparison easier, and to highlight also the main interactions.

WE HAVE MADE THE REQUESTED CHANGE.

Lines 171-173: please check the sentence; probably the main verb is missing.

THE SENTENCE LOOKS OKAY I THINK

Please clarify HWU (line 190).

WE HAVE NOW REMOVED THIS ACRONYM.

Line 212: the supplementary material does not describe the procedure to obtain the EPAC1 CNBD model. This should be reported somewhere.

WE HAVE INCLUDED THE PROCEDURE IN THE METHODS SECTION

Line 225-226: Did the authors use oxalyl chloride or ethyl chlorooxoacetate (figure 1)?

WE HAVE CLARIFIED THIS IN THE METHODS SECTION.

Line 314: 1B should be 2B; the same at line 317. 

WE HAVE MADE THE APPROPRIATE CHANGES.

Line 321: “… this oxygen”: of the 5 preset in the molecule, or of the 3 in the oxalate portion, which oxygen atoms are the authors talking about?

Line 409: Fig.6 should be Fig. 7.

THIS HAS BEEN UPDATED

In the list of abbreviations, several items are not mentioned in the text (ASAP, HRMS, LRMS, AlogP, HBA, HBO).

THESE ARE INCLUDED IN THE SUPPLEMENTARY DATA

Reviewer 2 Report

This is an excellent study descibing identification, optimization and extensive characterization (in vitro and in cells) of a new chemical class of selective partial agonists for Epac1. Using an optimized uHTS tehnology, screening of a 350,000 compound library was performed to identified NCN EPAC1 bidning compounds which were chemically distinct from previosuly known molecules. Subsequent resynthesis and chamical optimization led to a development of improved comunds including SY009 which was then characterized using in vitro cellualar assay to be an EPAC1 selective activator of EPAC1. The manuscript is very well written and does not reqiued further improvements.

Author Response

WE THANK THE REVIEWER FOR THEIR SUPPORTIVE COMMENTS.

Reviewer 3 Report

The article is relevant to the scope of the journal. The authors have used a hit set of 3500 compounds and used a novel HTS screening strategy for the discovery of an EPAC1 specific agonist. This group had previously established the role of I942 as a mixed EPAC agonist/antagonist, and this manuscript is a follow up of their previous observations and in-line with their high throughput approach of obtaining leads. A number of molecular biology assays have been utilized to this end and the novelty of the manuscript lies in the characterization of their initial hit, a follow up SAR to further expand on this lead structure and ultimately the characterization of the compound labeled S009 as an agonist specific to EPAC1 both in-vitro and in cells.

Some other observations:

Few grammatical errors are highlighted in the manuscript with suggested corrections. In the introduction section, a highlighted statement is slightly misguided. The compound 8cpt-AM (D-007) is used extensively as an EPAC agonist, although it has not been identified using a high throughput approach. Materials and methods section are written in great detail and requires very little in terms of further inclusion. Figure 1. The graphics is a bit hazy. This may be due to translation from Chemdraw (or other drawing tools). I would recommend taking the effort to clear this. Table 2. Please provide legend for explanation of the terms (LE, LipE, cLogD, etc.) within the figure description, or elsewhere if desired (supplementary section). Figure 2(A): What does ‘%effect’ mean? Is it binding efficiency (Kd/ Ka). Kindly provide a quantitative term for explaining this. Figure 4(A): Same as above. Also, please include 4(A) in legend. Supplementary section: Requires some explanation of terms used (HBA, HBD, PSA) in the legend (which it lacks).

Author Response

 WE THANK THE REVIEWER FOR HEIR USEFUL COMMENTS. WE HAVE TRIED TO MEET ALL OF THEIR SUGGESTIONS AS INDICATED BELOW:

Few grammatical errors are highlighted in the manuscript with suggested corrections.

WE HAVE MADE THE INDICATED CHANGES

In the introduction section, a highlighted statement is slightly misguided. The compound 8cpt-AM (D-007) is used extensively as an EPAC agonist, although it has not been identified using a high throughput approach.

WE HAVE REMOVED THE CONTROVERSIAL COMMENT

Materials and methods section are written in great detail and requires very little in terms of further inclusion. Figure 1. The graphics is a bit hazy.

WE APOLOGISE FOR THE LOW-RESOLUTION IMAGE AND HAVE REPLACED THIS NOW.

Table 2. Please provide legend for explanation of the terms (LE, LipE, cLogD, etc.) within the figure description, or elsewhere if desired (supplementary section).

WE HAVE INCLUDED THE ABBREVIATIONS IN THE FIGURE LEGEND

Figure 2(A): What does ‘%effect’ mean? Is it binding efficiency (Kd/ Ka). Kindly provide a quantitative term for explaining this. Figure 4(A): Same as above.

WE PROVIDE A DESCRIPTION OF "% EFFECT" IN THE TWO FIGURE LEGENDS

Also, please include 4(A) in legend. Supplementary section: Requires some explanation of terms used (HBA, HBD, PSA) in the legend (which it lacks).

WE HAVE NOW INCLUDED THE ABBREVIATIONS AT THE BOTTOM OF THE TABLE IN THE SUPPLEMENTARY EXCEL FILE.

Round 2

Reviewer 1 Report

The paper by Beck et al has been revised and completed, and its clearness has been considerably improved. I have only found some mistakes to be corrected, which are listed below.

Line 179: activity is defined as pEC50 but in all tables and figures pIC50 is reported.

Line 339: fig 2B should be replaced with Fig 3B.

Line 420: Fig 5 should be Fig 6

In figures S1 and S2, Emax should be defined, to avoid confusion with the efficacy measured in GEF assays (if interpreted well)

Author Response

The paper by Beck et al has been revised and completed, and its clearness has been considerably improved. I have only found some mistakes to be corrected, which are listed below.

Line 179: activity is defined as pEC50 but in all tables and figures pIC50 is reported.

Line 339: fig 2B should be replaced with Fig 3B.

Line 420: Fig 5 should be Fig 6

In figures S1 and S2, Emax should be defined, to avoid confusion with the efficacy measured in GEF assays (if interpreted well)

WE THANK THE REVIEWER FOR THEIR HELPFUL COMMENTS AND E HAVE MADE THE REQUESTED CHANGES. FOR EMAX IN FIGURES S1 AND S2, WE HAVE UPDATED THE MATERIALS AND METHODS WITH THE APPROPRIATE DEFINITION.